# Data Experts as the Balancing Power of Big Data Ethics

Richard Novak and Antonin Pavlicek *

Department of Systems Analysis, Faculty of Informatics and Statistics, Prague University of Economics and Business, 130 67 Prague, Czech Republic; richard.novak@vse.cz
* Correspondence: antonin.pavlicek@vse.cz; Tel.: +420-224-09-5473

**Abstract:** In this theoretical paper, we explore Big Data ethics in the broader context of general data ethics, stakeholder groups, demand for governance and regulation, social norms, and human values. We follow and expand on the digital divide, governance, and regulatory theories, and we apply them to many levels and contexts, such as state and society, organization, enterprise governance of IT (EGIT), and data projects, among others. We introduce the new role and responsibility of data experts as an important stakeholder group in the balance of power of Big Data ethics because they simultaneously hold a position in groups of data-rich organizations and data-poor users. We argue that the balancing role of data experts consists of motivation and competence, a sense of responsibility for data ethics, and the possibility and means to influence Big Data issues. Finally, we conclude our research by model mapping the role of data experts in Big Data ethics and proposing them as a balancing power.

**Keywords:** big data ethics; balancing power; data experts; data-rich; data-poor; governance; regulation; social norms

## 1. Introduction

In this paper, we explore Big Data ethics in the broader context of general data ethics, different stakeholder groups, demand for governance and regulation, social norms, and human values. We introduce the new role of data experts as an important stakeholder group that could be a balancing power of Big Data ethics, because they tend to simultaneously hold a position in data-rich organizations and are personally data-poor users.

We argue that the power of data experts is derived from their background knowledge of the positives of Big Data use cases but also from the related negative issues (e.g., [1,2]). Their responsibility stems from the common identity produced through membership in professional groups regarding data ethics [3].

The motivation for this paper is to shift the discussion about data ethics from the current focus on privacy intrusion to more general issues, such as the digital divide and the balance of power in society. In related works (e.g., [4,5]), this movement of data ethics priorities is already visible. However, the majority of papers still focus on an analytical description of Big Data ethics issues. We see our contribution in the creative part of the paper where we suggest a possible solution for digital inequalities in a new balanced model that respects the different roles and responsibilities of all involved parties (organizations, users, and experts).

We discuss Big Data as a socio-technological phenomenon rather than only a technological one [6]; examine the specifics of Big Data compared to general data ethics [1,7]; compare conservative, liberal, and digital approaches on how to govern and regulate society [8–11]; and describe social norms and respected human values [12,13] as an important part of the principles of governmentality [10].

Our approach to Big Data ethics spans from the computer ethics described by Wiener [14] and Moor [15] to the modern information and data ethics formulated by Capurro [16] and Floridi and Taddeo [7], and, finally, to the specifics of Big Data.

The specifics of Big Data that we describe are the following: the specific role of stakeholder groups, use cases of Big Data showing mainly the positive benefits, demand for governance and regulatory frameworks, conflicts, and issues stemming from the clash between Big Data use cases and ethics (see [1]).

In our paper, we investigate more deeply the interplay of the specific role of stakeholder groups in relation to the rising demand for governance and regulatory frameworks.

Based on the research on conservative, liberal, and digital approaches to governance at different levels, we continue to work with the existing regulatory and governance frameworks at different levels, such as the state and society, organization, and IT and data projects, among others. Following Harvard law professor Lawrence Lessig [8], who named four different areas for how society can be regulated and governed (market, law, social norms, and architecture), we further research the power of social norms. Although we focus mainly on social norms and human values, we expand on Michel Foucault's concept of governmentality [10] to the specifics of the current data-driven society.

We highlight the roles and responsibilities of data experts registered in various professional associations (both national and international) who should balance the conflict between data-rich (organizations) and data-poor (users) groups.

We conclude the paper with Figure 1, which proposes a model mapping the role of data experts in Big Data ethics and describing them as a balancing power. This model is based on the digital divide theories and the principles of governance and regulation that we develop here.

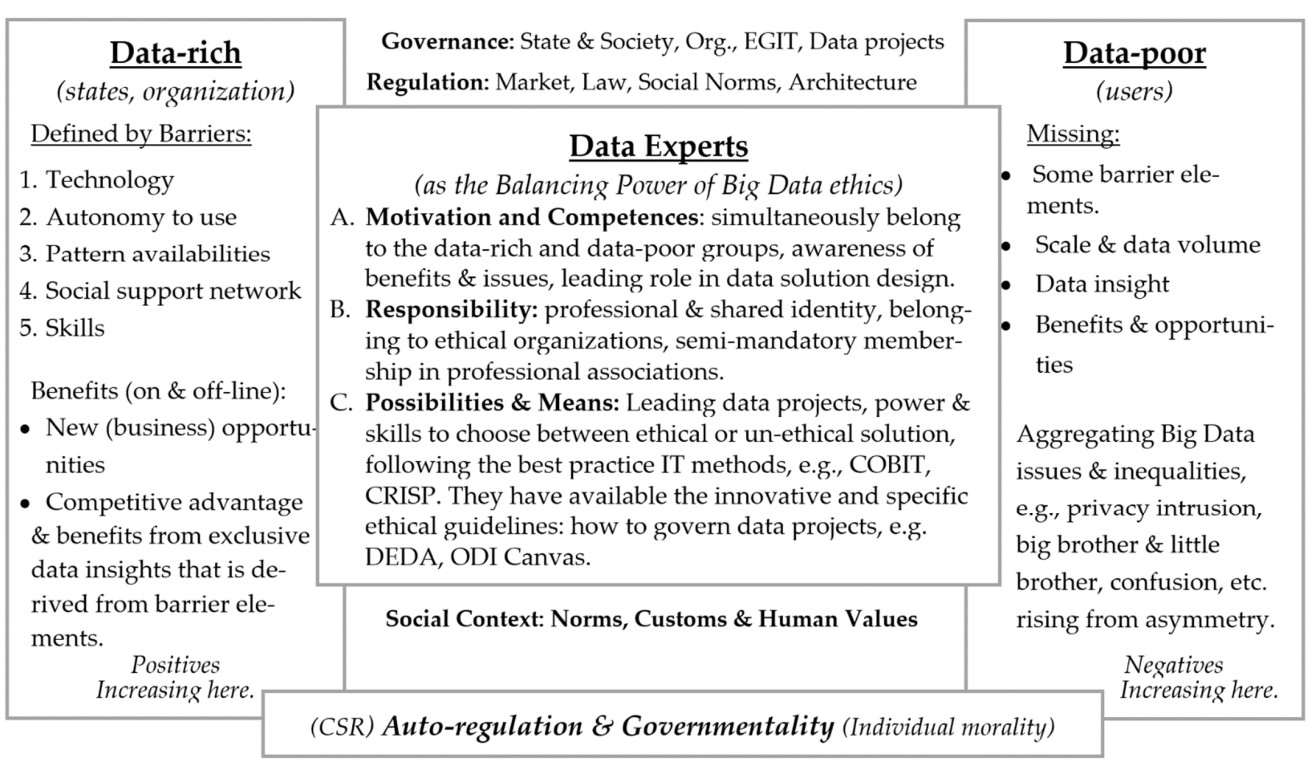

**Figure 1.** Model mapping the role of data experts in Big Data ethics and describing them as a balancing power.

Finally, we discuss this concept with some parallel concepts and argue why this proposal could be more effective than other means of regulation and governance, such as the market, law, or architecture.

The main contributions of this paper are identifying the leading stakeholders and their interest in the Big Data era; shifting the discussion from privacy to power issues; explaining that the social responsibility strategy (CSR) in data-rich organizations is not sufficient to balance the power; formulating the role of data experts as the balancing power

in data ethics; and proposing the model in Figure 1 as a new device, which shows this balancing power in the context of different theories.

## 2. Materials and Methods

This paper is a theoretical overview and focuses on an interdisciplinary approach mainly in the fields of computer science, philosophy, and sociology. First, we conducted a literature review, and all the original resources are named and listed in the References. Our literature research follows the PRISMA scheme [17]. The records identified (both through database searching and from other sources) n = 125. We removed 76 sources through a screening process. Forty-nine full-text articles were assessed for eligibility. Thirty-two number resources were included in the synthesis and the article.

The theoretical research was based on logical methods such as analysis vs. synthesis, induction vs. deduction, and abstraction vs. concretization, but we primarily used the design science methodology approach [18], with the intent to achieve a better understanding of the problem domains of Big Data and ethics by the construction of the designed device "Model mapping the role of data experts in Big Data ethics and describing them as a balancing power" shown in Figure 1. The final device enables us and the readers to better grasp the problem and is a new contribution of the authors.

## 3. Results

### 3.1. Big Data Ethics in a Broader Context

The research conducted by other authors shows that there has been a long-term discussion about the impacts of new technologies on society and their ethical norms and values. Computer ethics have been evolving since the invention of computers in the 20th century after the world wars and has been described by many authors, such as Norbert Wiener [14], Walter Maner [19], or James Moor [15]. However, the foundation of modern information ethics was established at the end of the 20th century by Rafael Capurro [16] and Luciano Floridi as described below.

Over the last few years, there has been a visible shift from information ethics to data ethics based on the idea of two Oxford academics, Luciano Floridi and Mariarosaria Taddeo, who stated the following: "We should concentrate on what is being handled (data) as the true invariant of our concerns and that is why labels such as 'robo-ethics' or 'machine ethics' miss the point." [7].

The most recent definition of data ethics is from 2016 and was provided by Floridi and Taddeo, approaching the topic on different levels of abstraction (LoA), such as macroethics, distinguishing the ethics of data, algorithms, and practices.

This respected definition of data ethics in LoA of data was set in their article "What is Data Ethics?" [7] as

> "*In the light of this change of LoA, data ethics can be defined as the branch of ethics that studies and evaluates moral problems related to data (including generation, recording, curation, processing, dissemination, sharing and use), algorithms (including artificial intelligence, artificial agents, machine learning and robots) and corresponding practices (including responsible innovation, programming, hacking and professional codes), in order to formulate and support morally good solutions (e.g., right conducts or right values). This means that the ethical challenges posed by data science can be mapped within the conceptual space delineated by three axes of research: the ethics of data, the ethics of algorithms, and the ethics of practices.*" [7]

#### 3.1.1. Specifics of Big Data Ethics

We do agree with the findings outlined in the latest work of Floridi and Taddeo; however, in our opinion, there are still some unnamed Big Data specifics:

1.  Different roles of stakeholder groups (organizations, users, and data experts);
2.  Possible use cases of Big Data (showing mainly positive benefits);
3.  Conflicts and issues stemming from Big Data benefits and human values;

4. Demand for governance and a regulatory framework.

In our paper, we provide an in-depth investigation into the interplay of the specific role of stakeholder groups in relation to the rising demand for governance and regulatory frameworks.

### 3.1.2. Description of Stakeholder Groups

The specific role of stakeholder groups means that there is increasing information asymmetry between the individual users that can be considered data-poor and big organizations (state organizations and corporations) that collect data about individual users and can be considered data-rich.

The term of information asymmetry is not new and is related to the phenomenon of the digital divide. The term digital divide in relation to IT technology was coined in the 1980s with the introduction of the Internet in America and its nonlinear spread between its first users.

Over time, the digital divide has been categorized into three parts: the first divide mainly regards the limited access to new technology [20], the second divide is more about the skills needed [21], and, finally, the third divide asks the question, "who benefits most from being online and then also in the offline world?" [22].

Thus, when speaking about stakeholders in Big Data, we can provide three important categories of stakeholder groups:

- data-rich organization;
- data-poor users;
- data experts as a balancing power.

To be data-rich brings insight into many areas of the economy and society and competitive advantages that generate new business opportunities.

The definition of data rich, based on the digital divide description, means having five barrier elements:

- Technical means, software, hardware, and connectivity quality;
- Autonomy of use (location of access, freedom to use the medium for one's preferred activities);
- Use patterns (types of uses of the internet);
- Social support networks (availability of others one can turn to for assistance with use and size of networks to encourage use);
- Skill (one's ability to use the medium effectively) [21].

Data-rich organizations nearly always have a corporate social responsibility strategy (CSR) in place that also covers the data ethics area. We could expect that for data corporations like Facebook, Amazon, and Google, among others, the CSR should be a good balancing power. However, many recently highly publicized affairs, e.g., the Cambridge Analytica scandal, show that it is not the case. The research of CSR and Big Data and analytics [23] showed that Big Data is underemployed in the area of corporate social responsibility. Their research and in-depth assessment conducted on a sample of the best-ranked global German companies, selected from the 2015 sustainability ranking reports, confirmed that these companies are not necessarily primarily interested in CSR but in economic interests.

A relevant question emerges: if organizations are not prioritizing CSR strategies, why would they heed the advice of a proposed data expert? Based on our proposal, data experts do not use their power in a way to increase the importance of CSR in organization priorities, but they can solve the ethical problem directly without involving management and their prioritization process. Data experts, as key people responsible for the design of Big Data solutions, can change an unethical solution to an ethical one by themselves when they are motivated and capable of such behavior.

Data poor means (opposite to data rich) missing one of the five barrier elements, namely, the technical means, autonomy to use them, ability to use patterns, access to social support networks, and possessing specific data skills.

However, even if the data user held all of above-mentioned elements, it is still mainly access to the critical volume of data that brings the competitive advantage hidden in data. Furthermore, despite it being highly unlikely that individuals would have all of these, even if they did, they would never have the capabilities of data-rich organizations on a comparable scale [4].

To be a data-poor user means providing personal data to data-rich organizations that benefit from it in the online and also offline world. Originally, it was a good trade-off, i.e., users exchanged their privacy for better personalized and often free services provided by organizations. However, over time, negative issues, such privacy intrusion, the big brother effect on the state level and the little brother on the corporate level, confusion, and others, have arisen and created inequalities affecting data-poor users. For a comprehensive insight into Big Data issues, see [5,24].

In addition to the data-rich organization and data-poor users that we identified, there is a third group of power users working in data-rich organizations that we call data experts.

A data expert is a member of both previously named groups: data-rich as part of a corporation and data-poor as an individual. In reference to Helen Nissenbaum and her concept of privacy as contextual integrity [2], we can assume data experts to be part of many contexts and therefore competent in Big Data ethics. This stems from their insight into the context of Big Data and awareness of many benefits and issues related to Big Data. The comprehensive insight originates from the leading role of data experts in the design of IT solutions that respond to business requirements assigned to them by organizations.

The possible special role of data experts is derived from the following three aspects:

A.　Motivation and competence;
B.　Sense of responsibility for data ethics;
C.　Possibility and means to influence Big Data issues.

The fact that data experts simultaneously belong to the data-rich and data-poor stakeholder groups ought to be a good motivation to be the third balancing power that mediates the opposite interests of both groups.

The responsibility related to the role of data experts could be derived from their belonging to a data special social group and professional and ethical organizations that group people based on a shared professional identity. Belonging to these groups is often confirmed by acceptance to official memberships in international, national, and corporate groups and associations.

> *"The common identity is produced and reproduced through occupational and professional socialization by means of shared educational backgrounds, professional training, and vocational experiences, and by the membership of professional associations (local, regional, national and international) and institutes where practitioners develop and maintain a shared work culture."* [3]

Membership in such a professional organization related to specific industries or occupations is usually not mandatory; however, there are some benefits, such as personal certification, training availability, access to knowledge bases, and possible participation in conferences that are available only to members of these organizations. In some cases, if you are not a member of such a professional organization, you practically cannot do your job, for example, doctors of medicine who are not members of Camera Medica (Medical Chamber).

We discuss and conclude the role of data experts and the possibility and means of influencing Big Data issues in Section 3.4, and Figure 1 provides a visual form of this conclusion.

### 3.2. Demand for Governance and Regulatory Frameworks

3.2.1. Governance Rules

There are many approaches to governance; however, governance as a general term is still considered vague and may have a different meaning in different contexts. Furthermore,

governance, as a steering principle, is overarching at many levels, such as states and society, organizations, information technologies, and data or security projects.

Comprehensive research on the theme of governance was provided by Petr Vymetal in the paper "Governance: Defining the Concept" [25]. As a summary of this research, we suggest using the following comprehensive definition that is appropriate for the majority of disciplines and levels:

> "*Governance is the system of values, policies, and institutions by which a society manages its economic, political, and social affairs through interactions within, and among the state, civil society, and private sectors. It is the way a society organizes itself to make and implement decisions—achieving mutual understanding, agreement, and action. It comprises the mechanisms and processes for citizens and groups to articulate their interests, mediate their differences and exercise their legal rights and obligations. It is the rules, institutions, and practices that set limits and provide incentives for individuals, organizations, and firms. Governance, including its social, political and economic dimensions, operates at every level of human enterprise, be it the household, village, municipality, nation, region or globe.*" [26]

In respect to Big Data ethics, we use the top-down cascading principle and find that the following governance areas are relevant:

- States and society governance;
- Enterprise Governance of IT (EGIT).

"States & Society Governance: A very comprehensive view" is provided by Bell and Hindmoor in their book, Rethinking Governance: The Centrality of the State in Modern Society (2009).

Bell and Hindmoor [9] name many different perspectives on state and society governance, such as a state-centric relational approach, hierarchy, top-down governance, governance through persuasion, governance through markets and contracts, governance through community engagement, and governance through associations.

In regard to the current data-driven changes that are supported by IT, namely, reducing hierarchy, adding complexity, and introducing new trends, and because of the new, open risks in digital public governance [11,27], it is very important to address the question mentioned also at the end of Bell and Hindmoor in "Rethinking Governance: How to govern society without Governance" [9].

This approach to state governance without governance was holistically described by the concept of governmentality and analyzed by Michel Foucault in 1978 [10]. He addresses the question of how conduct governance in the emerging global society. He describes his observation of the contemporary population that can be governed by a government through apparatuses of security using a political economy. The security apparatuses are to provide the population with a general feeling of well-being. In doing so, Foucault recommends that the state's actions be rather restricted and subtle in their nature, yet consequently very influential in their outcome. He offers a depiction of a multicentered society, which, even from the position of a state, is best regulated by the market mechanisms and by injecting the individuals with ideas rather than forcing on them the government's will by force. The individuals then become auto-regulated and auto-disciplined.

In order to be able to govern in this manner, to conduct such governmentality, an extensive amount of information about the population is required. With the development of Big Data, this becomes increasingly accessible. The problem is that it is not primarily the state gathering the information; it is the technology companies and, subsequently, other business sectors in the market.

Faced with the new phenomenon of the global technology companies, it may be rather difficult for the states to control them. In any case, the answer to these questions is not to open a vast conflict of sovereignty and discipline between the "private" Big Data and the government. On the contrary, for states, it should prove more effective to engage with the technology companies using the instruments of Foucault's governmentality as

described above. This temptation to use commercial data collected by corporations about their clients for the purpose of the state to control their inhabitants came true recently in China. The Chinese government introduced their social credit score system, scoring the behavior of each individual in China, and in 2018, approached the Alibaba e-commerce platform for the initial data to calibrate their model. This system, which has been in full operation since 2020, is based not only on the payment history of individuals but also on the monitored behavior of individuals [28,29]. When discussing the mechanism of an autoregulated society wherein Big Data is an essential component of this governance, which is our extension of Foucault's observation, we should better describe the other forces of the regulation mentioned above, ideally by some widely accepted frameworks that we briefly introduce.

For the level of state, society, and organizations (covering also enterprises), we can consider the law as this widely accepted framework that we describe further. For the area of IT in enterprises, where the data experts operate, "IT best practice" standards are more respected by these data experts than the law. Enterprise governance of IT (EGIT) is an important part of the governance of organizations (corporations/enterprises or even state organizations).

Several institutions are concentrated on research and development in the area of EGIT. The most important seems to be the IT Governance Institute (ITGI), which is a branch of the ISACA (Information Systems Audit and Control Association), an independent, nonprofit global association.

The relevance of ITGI is essential because of its connection to COBIT (Control Objectives for Information and Related Technology), which is a standard also supported by the ISACA. Currently, COBIT 5 is considered the most comprehensive framework in the area of IT governance and management.

> "*Enterprise governance of IT (EGIT) is defined as an integral part of enterprise governance, exercised by the Board, overseeing the definition and implementation of processes, structures, and relational mechanisms in the organization that enable both business and IT people to execute their responsibility in support of business /IT alignment and the creation of business value from IT-enabled business investment.*" [27]

We have no space in this paper further discuss EGIT and COBIT; however, we want to highlight here their relevance for data experts. In the data ethics area, it is important that the same cascading and steering principle of governance described above are applied to different levels, such as states and society, organizations, information technologies, and finally also to data projects.

For our approach to Big Data ethics and the model introduced in Figure 1, the following terms are important: the role of stakeholders, balancing powers, principles, rules, and motivations (goals), among others. They exist in all relevant governance models even though their meaning can be slightly different depending on the governance level. What is essential for the role of data experts as individuals is that they are part of many governance models, and all of these governance models are logically interconnected (cascaded) in a global society.

### 3.2.2. Regulatory Framework

The American lawyer and respected professor of law at Harvard University, Lawrence Lessig, in his book "Code and Other Laws of Cyberspace" [8], described several factors as the need to regulate cyberspace, such as the market, legislation, social norms, and architecture.

> "*Our choice is not between "regulation" and "no regulation." The code regulates. It implements values, or not. It enables freedoms or disables them. It protects privacy or promotes monitoring. People choose how the code does these things. People write the code. Thus, the choice is not whether people will decide how cyberspace regulates. People–coders–will.*" [8]

Although in computer and data science, "code" typically refers to the source code of a computer program, in law, "code" usually refers to valid legislation. In his work, Lawrence Lessig explores how code in both senses serves as an instrument for social control, leading to his maxim that "code is law."

With rapidly developing technologies, the ever-growing generation of data, the increasing power of corporations, and noting that the law (code) usually works retrospectively, the importance of data experts as "coders" that write the new rules is currently essential.

We do not have enough space to describe all of the above-named forces in detail; thus, the four Lessig forces must suffice here, and we comment in-depth on the forces of social norms and human values. We consider them to be the most relevant to the Big Data ethics discussed in this paper.

For Lessig, the market means the general market principles governing a society.

The law means the whole legislative framework, such as constitutional law, civil codes, corporate law, and criminal codes, where some special legislations, such as GDPR (General Data Protection Regulation), are very relevant for Big Data ethics in the European Union and also in California, USA.

The term architecture was used originally by Lessig as one of the four general powers regulating social systems; thus, it is not equivalent to the commonly used term for IT architecture, e.g., in the TOGAF framework. However, Lessig's architecture is used as a possible equivalent to the software code that is similar to Big Data solutions created by data experts discussed in this paper.

Social norms and human values are an important part of ethics as practical philosophy and guide the behavior of individuals in stakeholder groups. As such, we provide an in-depth discussion of this in the next section.

### 3.3. Social Norms and Human Values

The data experts hold the position and role as employees of organizations and simultaneously as users. In both roles, they are acting individually and make decisions on their own; however, to decide as an individual takes into account many factors that are not visible at first glance.

The philosopher Jan Sokol [30], building on the work of other philosophers, introduces three different sets of rules on how individuals govern themselves in society and its social norms:

- Social custom;
- Individual morality;
- Ethics.

Social custom regards cultural stereotypes that are earned unconsciously and automatically like we were taught as children.

Individual morality means morality as a voluntary self-restriction, regardless of the actions of the majority or the state of a given society.

Sokol argues ethics as the best way to regulate relations in a society: "ethics as a search for what is best" [30]. This accounts for positive action of an individual, a creation. It involves continuously asking the following question: what is right and what is wrong? Furthermore, it also necessary to be motivated and able to search for new answers in new contexts.

Regarding the relationship between human values and social norms, we follow leading sociologists from the previous century, such as Emile Durkheim (1897–1964) and Max Weber (1905–1958), who stated that human values are a central concept for explaining the social behavior of groups and individuals. As stated by the Israeli sociologist Shalom H. Schwartz,

> "*Values have played an important role not only in sociology, but in psychology, anthropology, and related disciplines as well. Values are used to characterize cultural groups, societies, and individuals, to trace change over time, and to explain the motivational bases of attitudes and behavior.*" [12]

In the next sections, we briefly introduce two complex concepts of human values, namely Schwartz theory of human values and the European Charter of Fundamental Rights, to identify the values that are generally accepted in a global society [12] and those also protected by law in the European Union.

### 3.3.1. Human Values Based on Schwartz Theory

Schwartz theory covers ten values recognized worldwide and has been confirmed by a comprehensive survey carried out on more than 60,000 individuals in 64 nations [12].

Although the ten values are recognized by all nations, they differ in their importance among different nations and individuals; thus, the values can be ordered by their importance relative to one another. It is also important to mention that these values can be considered as beliefs that are impossible to separate from actions.

It is important to consider that these ten values interact together, and we can also group them into the two large groups of five based on whether they have either a personal or social focus.

The five values with a personal focus are achievements, power, hedonism, stimulation, and self-direction.

The five values with a social focus are security, conformity, tradition, universalism, and benevolence.

### 3.3.2. Human Values Based on EU Charter of Fundamental Rights

The declaration of human rights has a long history worldwide, especially in the US and Europe after the second world war.

In 2000, the Council and the Commission of the European Parliament [13] solemnly proclaimed the Charter of Fundamental Rights of the European Union. The following list of values can be derived from this document:

- Dignity means human dignity, a right to life, the integrity of a person, etc.
- Freedoms mean the right to receive and impart information, liberty, and security, respect for private life, protection of personal data, etc.
- Equality means equality before the law; cultural, religious, and linguistic diversity; equality between women and men; etc.
- Solidarity applies to workers' rights, the prohibition of child labor, family protection, social security, healthcare, etc.
- Citizen rights mean the right to vote and run for political office, the right to good administration, freedom of movement and residence, etc.
- Justice means the presumption of innocence and right of defense, etc.

In the following section, we discuss the importance of human values in the role of data experts as the balancing power of Big Data ethics.

### 3.3.3. Social Norms, Human Values, and Data Experts

As discussed earlier, data experts simultaneously hold positions in groups of data-rich organizations and data-poor users.

Jan Sokol, who formulated the three distinctive approaches to social norms described above (social customs, individual morality, and ethics), also discusses in his book "Ethics, Life and Institutions" [30] the role and responsibility of individuals that are members of institutions (organizations) and that take part in their decision processes.

For the role of data experts, it is essential to emphasize that no matter the situation in which we find ourselves today in the field of data science and the globalized world, we are acting and responsible as individual people. This means that the social norms and human values should serve data experts as the essential guidance system working in both stakeholder groups.

### 3.4. Data Experts as the Balancing Power of Data Ethics

In the section dedicated to stakeholder groups, we described the special role of data experts that is based, in our opinion, on three elements: motivation and competence; sense of responsibility for data ethics; and possibility and means of influencing Big Data issues.

The motivation of data experts arises from the fact that they simultaneously belong to the data-rich and data-poor stakeholder groups. The competence comes from the insight of data experts into the context of Big Data and the awareness of the positive benefits of use cases practiced by data-rich organizations, as well as the negative issues impacting mainly data-poor users.

The responsibility comes from their shared professional identity and their belonging to a special social data group and professional and ethical organizations. Official membership in the professional associations is not mandatory; however, it brings many benefits, and we can expect the increasing importance of such associations for practicing the job of data experts in a similar manner to how doctors of medicine are forced to be members of Camera Medica.

The possibility and means of influencing Big Data issues refers to the means of data experts to influence Big Data issues arising from the three following aspects:

- The leading role of experts in data projects;
- Respecting the IT-relevant, best-practice methods;
- Using a data ethics guidance system to support ethical thinking.

The leading role of experts in data projects means that IT projects are typically managed in an organization based on the distribution of powers, roles, and competencies in project teams. Project governance and management follow certain best-practice standards, e.g., Prince, PMI, ITIL, and others. Project governance respects the roles in the team, e.g., business owner, solution designer, project coordinator, etc.

The only comprehensive insight into a data project belongs to data experts that need to understand both the business domain and the technical details of the solution that resolve the problem of the business domain.

As a business domain, we understand verticals, such as finance, retail, and manufacturing among others, and also horizontal domains, such as marketing, sales, operation, and accounting among others.

As technical domains, we identify many different areas, such as computing, networking, databases, and many others, or more general hardware, software, services and processes among others.

The leading role of data experts originates from the unique role in the design of IT solutions that solve, in many different and creative ways, the business requirements assigned to them by organizations.

Respecting the IT-relevant, best-practice methods includes, apart from the general project management methods mentioned above, data-specific methods, such as software engineering and pattern recognition, agile methods, architecture and solution design methods, and many others.

Regarding the term governance, we should mention COBIT 5, DAMA-DMBOK, COSO, ITIL, and ISO/IEC 27 0xx. Concerning the process of solution design, the CRISP-DM (Cross-Industry Standard Process for Data Mining) is usually a very relevant method for data experts.

Becoming a data expert is usually a long career path that starts at university and is achieved through practice to become a true data scientist with a wide range of experience and a leading role in writing the "code of data solutions and Big Data ethics".

Using a data ethics guidance system to support ethical thinking has not yet been well-described. Usually, there is a data etiquette, or "netiquette" (rules regarding how to behave in the online world), that is part of the CSR at an organization.

Following the belief "ethics as that the best to regulate relations in society" [30], data experts appreciate guidelines that support the process of continuously asking the following question: what is right and what is wrong?

As part of the research for this article, we explored the Data Ethics Canvas produced by the Open Data Institute (ODI) and DEDA methodology developed at Utrecht University. Although the ODI Canvas can be helpful in providing a set of essential areas to consider when working with data, the DEDA framework goes further in a detailed process and is a possible guideline for data experts.

DEDA supports continuously asking what is right and what is wrong. DEDA is an abbreviation for Data Ethics Decision Aid. It is an ethical assurance approach focused on Big Data projects that are based on a guided discussion that should include all of the people relevant to a project and take place before the Big Data system is designed.

DEDA implementation consists of three steps: learning methodology, organizing the project with all stakeholders, and asking a set of 29 predefined ethical questions relevant to data governance. A few examples of DEDA predefined ethical questions are as follows: Is there someone in the team who can explain how the algorithms in use work? Can you communicate how the algorithms work? Where do the data(set) come from? [31]. The DEDA framework was improved in an iterative process (2016–2018) and has since been applied by various Dutch municipalities. We consider DEDA and its process as a well-documented test subject about data ethics and a value-sensitive design approach applied to data projects in organizations that clearly declare their values, as described by Franzke et.al in 2021 [32].

Our model shown in Figure 1 does not provide detailed guidelines in the decision-making process of data projects like DEDA but describes the roles and responsibilities of different stakeholders in the context of different regulatory and governance frameworks and appeals to data experts to use their balancing power in a divided society.

To summarize this section, the balancing role of data experts involves three main aspects: motivation and competence; sense of responsibility for data ethics; and possibility and means of influencing Big Data issues. Furthermore, we argue that a data expert is a unique role in which all three of the above-mentioned aspects are possessed, and, therefore, it could be the balancing power of data ethics.

## 4. Discussion

In this paper, we formulated the new role and responsibility of data experts as a balancing power of Big Data ethics. We provided a follow-up and expanded on the digital divide, governance, and regulatory theories, and we applied them at various levels and in contexts.

The digital divide theories [20–22] support our approach to the two categories of data-rich organizations and data-poor users. Based on the research of conservative, liberal, and digital approaches to society governance [9–11,27], we continue to work with the existing relevant frameworks. We explored a cascade of governance at different levels, such as the state and society, organization, and IT and data projects among others, and we conducted an in-depth examination of the enterprise governance of IT (EGIT).

Following the research of regulatory frameworks [8], we further investigated the social norms and human values as guiding principles for data experts in different roles [30].

We suggest that the balancing role of data experts involves three main elements: motivation and competence; the sense of responsibility for data ethics; and possibilities and means of influencing Big Data issues. Additionally, we argue that a data expert is a unique role that possesses all of these elements and, therefore, could be a balancing power of Big Data ethics.

Finally, we conclude the research of other authors and our own contribution with Figure 1, which proposes a model mapping the role of data experts in Big Data ethics and describing them as a balancing power.

Even though this paper is theoretical research that expands on many resources and therefore opens a theoretical discussion, we encourage the debate of whether we should prefer the more liberal approach of Michel Foucault and his analyzed concept of govern-mentality (1978) [10] that we adapted to our digital society or whether some more strict or

conservative approaches like legislation or the COBIT framework among others should prevail.

We think that besides the theoretical discussion, our study touches on the practical implication concerning organizations and their mainly declarative role of CSR. A discussion regarding professional data associations and their enforcement of data etiquette from their mandatory/semi-mandatory members should also be held.

## 5. Conclusions

The results obtained in the paper are the following: there is an accelerating inequality phenomenon in the Big Data era stemming from the three digital divides; data-rich organizations are gaining enormous power, and their CSR strategies are not a guarantee of their ethical behavior. The data-poor users are aggregating the negative issues while providing benefits to data-rich organizations at the same time. We introduce the new role and responsibility of data experts as an important stakeholder group that could be a balancing power of Big Data ethics because they simultaneously hold the position in groups of data-rich organizations and data-poor users. The model shown in Figure 1 is the new device of our research and describes the role of data experts in the contexts of different Digital Divide theories and regulatory and governance frameworks, and it also describes their motivation and competence, sense of responsibility regarding data ethics, and possibility and means of influencing Big Data issues.

The limitations of this research are the missing quantitative and qualitative surveys supporting the theoretical aspects of the paper. We used a literature review and data here as a source of arguments for our model; however, we must consider these arguments as secondary and not primary data. It is important to supplement this conceptual paper with a survey among the data experts and associations about their awareness of the urgency regarding the digital divide and their readiness to take over responsibility as a balancing power of Big Data ethics.

As a subject for further research, we suggest the above-mentioned survey and also more in-depth study of the area of possibilities and means of influencing Big Data issues.

**Author Contributions:** Conceptualization and writing, review, editing, visualization, and original draft preparation: R.N.; resources, methodology, editing, and supervision: A.P. All authors have read and agreed to the published version of the manuscript.

**Funding:** This research received no external funding.

**Acknowledgments:** The paper was processed with a contribution of grant IGS 27/2021 from the Faculty of Informatics and Statistics, Prague University of Economics and Business.

**Conflicts of Interest:** The authors declare no conflict of interest.

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
