# Peer review of "Data Experts as the Balancing Power of Big Data Ethics"

_information, doi:10.3390/info12030097_

Round 1

Reviewer 1 Report

Thank you for the opportunity to read and review this paper. As I understand it, the main argument being made is the introduction of the new concept of "Data Expert" which is intended to develop a position that will bring ethical perspectives to the decision-making process for organizations.

Overall, I think this is an interesting and important topic, and the manuscript does a good job of engaging with relevant literature related to the topic. The topic is certainly of interest to readers, though could use further development. In general, significant language editing is needed -- the current grammatical structure makes the paper difficult to fully understand at parts. For example, in just the abstract, there are several issues. On line to "applied" should be "apply' and "are" should be "as". On line 15, I believe "influent" should be corrected to "influence" and "we argue these things" is unclear. An extra "the' before Model mapping on line 16 creates confusion. Likewise, editing throughout the manuscript is needed, including citation style inconsistencies. Additionally, some words appear often, but not always capitalized, such as Data. It is not clear what pattern such capitalization is following and should be double-checked and clarified. Sometimes other sections of the paper seem to be referred to as other chapters -- or perhaps this is a chapter that was converted into a manuscript? Either way, this language needs to be updated.

One major tension in the paper is that the author claims on p. 4 that companies are not interested in the CSR strategy in light of the pursuit of profit, so it's not clear why Data Experts would be given any particular weight by such organizations either. For me, this is the single biggest theoretical tension that must be addressed and resolved in the paper. To say this another way: if organizations aren't prioritizing CSR strategy, why would they heed the advice of a proposed Data Expert? Additionally, on line 151, it says the research paper has confirmed companies' focus on economic interests, but it is not clear how this was confirmed. 

Lines 160-165 note negative issues like privacy intrusion and the big brother effect, but might benefit from mentioning the corporate angle of "little brother" using a source such as “Little Brother: How Big Data Necessitates an Ethical Shift from Privacy to Power.” In Booth, P. and Davisson, A. (Eds.) Controversies in Digital Ethics (pp. 13-28). New York, NY: Bloomsbury.

Theoretically, the description of Foucault's governmentality framework needs some revision. The current wording in the paper states that Foucault suggests a particular approach for governments, but Foucault is analyzing this shift rather than endorsing it. This should be clarified in the paper. Relatedly, the author argues that "for states, it should prove more effective to engage with technology companies using the instruments of Foucault's governmentality..." This claim needs elaboration, as it is not clear exactly what actions the author is recommending. 

Immediately after that sentence, there is an abrupt transition to a discussion of enterprise governance of IT. This jump from one topic to another was jarring, and it was not immediately clear to this reader why such a large jump was made or how these topics were supposed to be related. 

Sections 3.3, 3.3.1, and 3.3.2 primarily summarize the works of other philosophers. While this might be helpful, these summaries are not clearly connected to the main argument of the paper. Why are summarizing these particular systems? Further, I would suggest that Wikipedia (p. 8) is not an appropriately credible source to cite in this context. This should be updated with a better citation. 

Finally, the article concludes with what seems to be a wholesale adoption of the DEDA framework. If this process is to be followed so closely, what distinguishes the proposed role of Data Expert from the DEDA framework? Why does the Data Expert need to be identified separately?

I hope this feedback can help strengthen this paper, as I believe this is a valuable line of inquiry and area of research that needs to continue to be explored. 

Reviewer 2 Report

Paper is theoretical. But authors considered an important problem of Big Data ethics.

Paper has practical value and elements of scientific novelty.

It has a logical structure, all necessary sections. 

Suggestions:

  1. Introduction section should be extended using more clearly the motivation of this paper. In the current form it is like Conclusion section
  2. It would be good to add point-by-point the main contributions in the end of the Introduction section
  3. It would be good to add the reminder of this paper
  4. It would be good to add Related works section 
  5. Authors declare that they research: "It is based on literature review and all original resources are named and listed in the reference chapter". You have used only 26 references. Where is the PRISMA scheme? How did you choose such references?
  6. The Materials and Methods section have only 3 paragraphs. Are you serious? This section should be significantly extended!
  7. Authors should add a Conclusion section. It should have: 1) results obtained in the paper; 2) limitations of this research; 3) prospects for the future research.
  8. A lot of references are outdated. Please fix it using 3-5 years old papers in high-impact journals.

Round 2

Reviewer 1 Report

The authors have addressed all of my concerns in their revisions. 

Reviewer 2 Report

Paper can be accepted.

It would be good to add non-itaretaive approach fo training when processing Big Data (DOI: 10.1007/978-3-319-91008-6_58)